# Chordoma Spontaneous Regression After COVID-19

**DOI:** 10.3390/v17010010

**Published:** 2024-12-25

**Authors:** Luis Fernando Moura da Silva Junior, Gyl Eanes Barros Silva, Marcos Adriano Garcia Campos, Antonio Augusto Lima Teixeira Júnior, Ramon Moura Santos, Orlando José dos Santos, Natalino Salgado Filho

**Affiliations:** 1University Hospital of UFMA, Federal University of Maranhao, São Luís 65080-805, Maranhão, Brazil; luisfernando@noz-neuro.com.br (L.F.M.d.S.J.); gyleanes@alumni.usp.br (G.E.B.S.); orlanddojs@hotmail.com (O.J.d.S.); natalinosalgadofilho@uol.com.br (N.S.F.); 2Duke Global Health Institute, Duke University, Durham, NC 27710, USA; 3Postgraduate Program in Genetics, Faculty of Medicine of Ribeirao Preto, University of São Paulo, Ribeirão Preto 14049-900, São Paulo, Brazil; aaltjr@usp.br; 4Dr. Carlos Macieira Hospital, São Luís 65075-441, Maranhão, Brazil; ramonmoura64@gmail.com

**Keywords:** spontaneous neoplastic regression, coronavirus infection, notochord, sarcoma

## Abstract

Chordomas are a low-to-intermediate-grade slow-growing subtype of sarcoma, but show propensity to grow and invade locally with recurrence and metastasis in 10–40% of cases. We describe the first case of spontaneous regression of a solid tumor (histologically and immunohistochemically proven chordoma) after COVID-19. A female patient with clival chordoma underwent occipitocervical fixation prior to tumor resection. In the early post-operative stage following the arthrodesis procedure, she was diagnosed with COVID-19. Six months after COVID-19, she finally came back for endoscopic endonasal resection of the tumor and pre-operative MRI surprisingly showed 98.9% regression of the tumor volume. Tumor resection was performed, and both histopathological and immunohistochemistry confirmed diagnosis of chordoma with positive brachyury levels. She showed improvement of right hemiparesis and left-sided tongue palsy. The tumor was comprised of tumor-infiltrating inflammatory cells. CD3 and CD68 were positive, suggesting the presence of T-lymphocytes and macrophages. CD20 and CD56 were negative, suggesting the absence of B-lymphocytes and NK-cells. The authors believe that the onset of COVID-19 exacerbated the patient’s immune response and improved anti-tumor immunity. It was concluded that T-cells, which are involved in the COVID-19 immune response and were found infiltrating the tumor, acted as a critical pathway to this event. Further studies are encouraged in order to gain a better understanding of the SARS-CoV-2–chordoma interaction.

## 1. Introduction

Chordomas are rare primary bone midline tumors that are thought to arise from remnants of embryonal notochord. Their incidence rate ranges from 0.18 to 0.84 per million people per year, showing variation across countries and racial groups [1]. They are very challenging tumors to treat and are considered to be a low-to-intermediate-grade slow-growing subtype of sarcoma. This information sometimes leads to the mistake that they are relatively benign tumors. They are aggressive and have a propensity to grow in a locally destructive and invasive way with a tendency to show local recurrence and metastasis in 10–40% of cases [2,3].

Currently, the median survival of patients is 7.7 years with a 5-year overall survival (OS) rate of 50–70%. When gross total resection with free margins is achieved, the 5-year OS rate increases to 65%; in inoperable cases, it decreases to 40% [2,3]. So, en bloc radical resection is the main treatment and poses various challenges, even for more experienced surgeons. Especially when the tumor is located in the skull base involving nerves and vessels, gross total resection (GTR) is often not possible without causing significant morbidity. In addition to GTR, radiotherapy and chemotherapy, many researchers have debated about the use of targeted immunotherapy.

Spontaneous tumor regression has been reported in the literature for a long time. In some cases, regression is related to concurrent infections. Four cases of spontaneous chordoma regression have been reported in the literature, but none are related to coronavirus disease (COVID-19) [4,5,6,7]. Some patients with hematologic malignancies supposedly demonstrated remission induced by severe acute respiratory syndrome coronavirus 2 (SARS-CoV-2) infection [8,9]. In this paper, we describe a rare case of spontaneous regression of a solid tumor and histologically and immunohistochemically proven chordoma after COVID-19 disease.

## 2. Case Report

A previously healthy 35-year-old woman came to our department after being referred for biopsy. She reported 9 months of a recurrent headache and 1 month of right hemiparesis (grade III) with a left-sided deviation of the tongue. Magnet resonance imaging (MRI) showed a 6.1 cm × 4.7 cm × 3.7 cm heterogeneous lesion in the anterior aspect of the craniocervical junction eroding the clivus, a left lateral mass of C1, and inferior portion of the odontoid process. It showed a huge extraosseous component protruding into the pre-pontine cistern, left cerebellum-medullary cistern, and foramen magnum. The lesion caused a mass effect over the pons, medulla oblongata, and upper portion of the cervical spinal cord, in addition to obliterating the left hypoglossal canal. Segments V3 and V4 of the left vertebral artery and V4 of the right vertebral artery were in direct contact with the tumor. Considering the image findings that were suggestive of chordoma with pain indicating craniocervical instability and the need to resect the C1 lateral mass and C2 odontoid process, we proposed occipitocervical fixation (C0–C2 fusion with lateral mass screws in C1 and pars screws in C2) and subsequent resection of the lesion (endonasal endoscopic approach) in 2–3 weeks.

After 6 months, she came back with the progression of symptoms. She had intense cervical pain, a headache and was unable to walk. MRI was repeated with the same finding of a 6.5 cm × 4.5 cm × 3.5 cm heterogeneous lesion suggestive of chordoma with a mass effect over the adjacent structures and bone invasion (Figure 1A). Physical examination revealed hyperreflexia and persistent left-sided tongue paralysis. On the day after hospital admission, the patient showed a decrease in consciousness levels. The lab tests showed leukocytosis, elevation in liver enzymes (AST and ALT), and c-reactive protein (CRP). Meropenem and amikacin were started. After three days, AST, ALT, and CRP levels started to decrease. Clinical status improved and she recovered her level of consciousness. A diagnosis of drug-induced hepatitis, related to the overuse of dexamethasone, painkillers and non-steroidal anti-inflammatory, was confirmed. After 14 days of clinical management, the patient’s results from the lab exams were back to normal, and surgery was confirmed.

In April 2020, she underwent the proposed spine procedure uneventfully. The patient’s cervical pain improved immediately in the post-operative period. A computed tomography (CT) scan showed adequate placement of the screws and rods. On post-operative day 3, the patient started feeling back pain in the cervical and dorsal region, which was refractory to routine analgesics and was initially attributed to myofascial pain secondary to hypomobility in bed. The pain progressed fast and the patient started presenting a fever and cough. The lab tests showed lymphopenia and an increase in the CRP level. The Chest CT showed bilateral ground-glass opacities. It was April 2020 and COVID-19 had just emerged with less than 20,000 cases in Brazil. A hypothesis of SARS-CoV-2 infection was considered and real-time polymerase chain reaction (RT-PCR) for SARS-CoV-2 test was performed with a positive result. Treatment was started with ceftriaxone, clindamycin, azithromycin, hydroxychloroquine, ivermectin, prednisone, zinc, and vitamin v. The patient had moderate symptoms with mild dyspnea requiring oxygen (O_2_) through a nasal catheter and clinical support.

After treatment and COVID-19 recovery, she decided to go back home and returned after the pandemic showing some improvement; therefore, tumor resection was performed. Even after the episode of COVID-19, she continued to report a headache and neuropathic pain for 5 months whilst taking gabapentin and amitriptyline.

At hospital admission in October 2020, a noticeable improvement in the patient’s motor and sensory function was observed. She could walk alone and her strength improved to grade IV. So, another MRI was performed to re-evaluate the tumor and define the surgical plan. A surprising regression of the tumor was noticed. A 1 cm × 0.6 cm × 0.5 cm lesion was restricted to the left side of the skull base, no extraosseous component was identified and there was no mass effect over the brainstem, cervical spinal cord, or even longus colli muscles (Figure 1B). Four hypotheses were considered: the lesion was inflammatory/rheumatologic and regressed after craniocervical fixation, an abscess that healed parallel to COVID-19 treatment with antibiotics, a lymphoma that regressed after the use of corticosteroids, or a spontaneous regression of the tumor. So, an endonasal endoscopic approach was taken to remove the residual tumor. The procedure was successful with GTR of the lesion and drilling the adjacent bone. Histological analysis was compatible with the chordoma (Figure 2A), which was confirmed by immunohistochemistry (IHC) showing positive reactions to panCK, EMA, Ki67 (<10%), S-100 protein and brachyury (Figure 2B). CD34, GFAP, and synaptophysin were negative. The patient showed continuous improvement in symptoms and achieved full recovery. Surveillance MRI was negative for both recurrence and residual tumor (Figure 1C). As association with COVID-19 occurred and SARS-CoV-2 screening of the specimen was performed with PCR and IHC. Both were negative.

Histopathological analysis also showed abundant tumor-infiltrating inflammatory cells. The leukocyte profile was defined by IHC as a cluster of differentiation (CD), including CD3 and CD68 positive cells, which suggests tumor-infiltrating T-lymphocytes and macrophages; CD20 was negative, suggesting absence of B-lymphocytes; and CD56 was scarce and rarely seen in inflammatory cells, suggesting the absence of natural-killer (NK) cells (Figure 3). Clear border limiting areas were observed between the necrotic area with an eosinophilic cytoplasm and the adjacent infiltrated area of tumoral inflammatory cells (Figure 4A–D).

At the patient follow-up in February 2021, she was able to independently maintain a KPS of 100 (Figure 5).

## 3. Discussion

We describe the first solid tissue with possible regression influenced by COVID-19. As chordomas are composed of notochord remnants, a very important differential diagnosis is ecchordosis physaliphora, which is a benign hamartomatous lesion, derived from notochord remnants. However, it is asymptomatic, usually occurs within the bone without extension to soft tissues, and also lacks contrast enhancement [5]. Our case showed a different presentation, which was not compatible with this diagnosis.

Following the report of our unique case, it becomes imperative to explore the broader implications of such an interaction. This case, marking a possibly unprecedented occurrence in solid tumors, prompts a closer examination of the mechanisms at play. Interestingly, this phenomenon is not isolated to chordomas alone. Pasin et al. in 2020 published one case of transient remission of lymphoma during the course of COVID-19. The tumor was composed of NK/T-cells, which are mainly suppressed (or even functionally exhausted) in the context of COVID-19. This promotes abnormal immune responses, which are less effective, and a cytokine release syndrome. While both diseases were active together, COVID-19 leads to normal NK/T-cell suppression/exhaustion, which are the cells that form the tumor, leading to remission. Even after recovering from COVID-19, patients may still experience lymphoma recurrence. [9].

In 2021, a 61-year-old man achieved remission of Hodgkin lymphoma after diagnosis of COVID-19. The authors attributed the event to an anti-tumor immune response triggered by SARS-CoV-2 [8]. A reduction in tumor size on imaging, decreased serum tumor marker levels, and improvement in oncological symptoms were also observed after SARS-CoV-2 infection, without a change in the therapeutic schedule, in some solid tumors such as lung adenocarcinoma [10], colon cancer [11,12] and renal cell carcinoma [13]. It is hypothesized that SARS-CoV-2 activates innate immunity by binding to the angiotensin-converting enzyme 2/neuropilin-1 complex on the surface of infected cell membranes via its spike proteins [14]. This interaction disrupts the natural tolerance of innate immunity through non-major histocompatibility complex restricted pathways, enhances the recognition of infected cells by macrophages and dendritic cells, triggers excessive inflammatory responses, and leads to the attack and destruction of infected cells by immune cells [14]. Additionally, there is cross-reactivity between SARS-CoV-2 antigens and specific tumor antigens that may activate tumor-specific T cells, leading to an anti-tumor response and oncolytic effects [8], and the activation of NK cells has been identified as a critical factor in promoting oncolysis [9].

Regarding chordomas, there are four cases of regression published (Table 1). In one case, the patient had just a suspected diagnosis since no biopsy was performed. This patient presented partial regression after dermatologic infection by *Mycobacterium marinum* with a 33% reduction in its volume after regression [6]. The other two cases had a confirmed diagnosis by histopathological analysis but none of them had positive brachyury. One had complete regression associated with post-biopsy *Escherichia coli* infection [4]. The other patient had no report of concomitant infection but it is possible that some medication, including herbal supplement and oils, could have triggered the 61.8% reduction in volume after regression [5]. In the most recently published case, not related to infection, diagnosis was confirmed by histopathological analysis and positive brachyury immunohistochemical staining. No measure of tumor volume was cited, only noticeable shrinkage of the tumor [7]. Our case showed the most considerable volume reduction of 98.9% after regression (Figure 1) and was both histopathologically and immunohistochemically proven to be chordoma (Figure 2).

The control of abnormal cell proliferation is an essential evolutionary requirement for mammals. The collective actions of the immune system, which involve both safeguarding the host and facilitating tumor development, throughout the oncogenic process, are referred to as tumor immunoediting [15]. This phenomenon allows the immune system to regulate and shape the cancer, resulting in negative selection. The process of tumor immunoediting consists of three distinct phases: elimination, equilibrium and escape.

The first phase, known as elimination, contemplates the recognition of tumor cells and their elimination (Figure 6) [16]. Initially, cells of the innate immune system are recruited through inflammatory signals induced by stromal remodeling, which signifies tissue damage. M1 macrophages are capable of capturing, phagocytizing, and lysing antigen-presenting tumor cells, thereby enhancing the cytotoxic capabilities of other immune cells, including CD8+ T cells and NK cells [17]. During this stage, natural killer (NK) cells are stimulated to produce interferon-gamma (IFN-γ), thereby promoting cell death. Additionally, certain cytokines are produced to hinder angiogenesis. Cell death occurs via apoptosis and the generation of reactive oxygen and nitrogen species. Eventually, tumor-specific CD4+ and CD8+ lymphocytes localize at the site of the lesion, leading to the destruction of tumor cells expressing the relevant antigens that still persist at the site. This phase occurs prior to the manifestation of clinical symptoms, and it is possible that a significant number of potential tumors are eradicated during this stage [15].

Tumor cells that exhibit variations and manage to survive the initial phase enter a state of equilibrium. During this phase, the remaining genetically unstable and mutated cells persist within an immunocompetent host, subjected to selective pressure from CD8+ T lymphocytes, IFN-γ, and IL-12p70-producing dendritic cells. This state of equilibrium prevents tumor progression and the formation of metastases [15,16].

In the escape phase, there is an anergy of protective effector cells and a prevalence of cells and cytokines exhibiting a tolerant profile. This escape occurs through various mechanisms, including the upregulation of key tolerogenic pathways, alterations in cellular proteins and receptors related to tumor antigen presentation due to mutations, modifications in the effector immune response, and dysfunction of the immune niche [16,18]. During this phase, tumor progression occurs.

Theoretical explanation of tumor regression usually considers trigger mechanisms that stimulate the host innate response to induce autoimmune immunotherapy against tumor cells, but defining these triggers remains a challenge.

A relevant factor to cite that could have influenced tumor regression was fever (Figure 7). Our patient presented episodes of high fever during the COVID-19 course. Tumor cells are more sensitive to heat than normal cells. Fever is observed in 25–80% of cases of tumoral regression and can have a role in this process. Following fever onset, the immune system increases its activity to carry out antigen recognition, being able to identify both infectious agents and tumor cells; increases the release of pro-inflammatory factors that are able to stimulate dendritic and T-cells [19,20]. Being more sensitive to heat, more cellular death will occur, and more tumoral cellular debris will express antigens that can activate antigen-presenting cells to produce a more powerful T-cell response [21].

Heat-Shock-Proteins, like HSP70, can participate in this process. They are not immunogenic per se but carry antigen peptides that trigger specific immunity to cancers [22].

With regard to the possible direct action of the virus itself in the tumor regression process, the screening of SARS-CoV-2 in tumoral cells was negative in both PCR and IHC techniques. It is crucial to understand that the possible COVID-19 mechanisms related to tumor regression do not arise from the direct presence of the virus inside tumor cells eliciting a viral-targeted immune response. Indeed, the influence should be indirect mainly via the improvement in the detection of tumor cells by reducing evasion tumoral strategies, improvement of the function of immune effectors directed to find and kill tumor cells and by cross-reaction with common antigens expressed by the tumor.

When acting as an oncolytic virus, SARS-CoV-2 may stimulate or refresh already existing but inefficient anti-tumor immunity or induce a novel non-self-antigen response. The virus, by enhancing antigen recognition and subsequent immune activation, stimulates a pro-inflammatory environment. In this context, it can withstand tolerogenic mechanisms aimed to facilitate viral infection to kill cells that are not protected by the immune system [16].

Even in the context of lymphopenia, especially with COVID-19, patients demonstrate hyperactivation of CD8+, with a correspondent overaggressive response, including high levels of expression of NK-cell-related markers and increased cytotoxicity [23]. Hyperactive CD4+ and CD8+ lymphocytes exhibit high concentrations of perforins and granulysin in cytotoxic granules [24].

During cytokine storms in patients with COVID-19, interleukin (IL)-6 and tumor necrosis factor (TNF)-α have been used to reduce suppressive functions of T regulator cells, resulting in an increase in IL-17 and reduction in interferon (IFN)-γ expression in inflamed tissue [25,26].

Another aspect that is not uniform among the published cases, but deserves attention, is the brachyury protein. It is the most suggestive finding in the diagnostic of chordomas. Immunogenic properties have been attributed to the brachyury protein. It is overexpressed in more than 95% of chordomas. Although it is located within the nucleus of the cell, CD8+ T cells specific to brachyury-bearing cell tumors have been characterized and lead to increasing intracellular expression of IFN, TNF, and IL-2. So, there is a possibility that brachyury expression contributes to a better immunoediting process and elimination of chordoma [27,28].

Furthermore, the brachyury protein is the target of a vaccine that aims to induce a highly efficient tumor-specific T cell response with increased activation of NK cells and cytokine production. A patient with metastatic sacral chordoma undergoing vaccine immunotherapy was diagnosed with COVID-19. Its course was uneventful, and the virus cleared rapidly. His medications included gabapentin, morphine, and methadone. At follow-up, he presented a fever and chills attributed to the vaccine infusion. He also had a significant improvement in pain related to the tumor. The authors suggest that maybe heterologous immunity with cross-protection, associated with innate immunity, might have contributed to SARS-CoV-2 rapid clearance in this case [29].

The occurrence of this cross-immunity between brachyury protein and SARS-CoV-2 suggested by Pastor et al. in 2020 may also have occurred in our case. In that case, immune response directed towards brachyury protein may have interacted with SARS-CoV-2 antigens, leading to a rapid clearance of the virus. But, in our case, perhaps this interaction happened in the opposite direction. We propose that the immune response directed towards SARS-CoV-2 could also have targeted brachyury-positive chordoma cells, optimizing the immunoediting process and tumor elimination.

The abundant tumor-infiltrating inflammatory cells observed in this case reinforce this possibility of a pro-inflammatory tumoral microenvironment with the presence of CD3+ T-cells and absence of CD20+ B-cells. CD56+ was scarce and this can be justified by NK T-cell suppression/exhaustion that occurs in the context of COVID-19. CD68+ showed considerable macrophage infiltration of the tumor, suggesting the process of elimination of the tumor probably by an immune-mediated process. The clear border limiting areas observed between the necrotic area and adjacent infiltrated area of tumoral inflammatory cells also support the hypothesis of an exacerbated immune process against tumors with an enhanced immunoediting process.

We believe that in our case, the onset of COVID-19 with exacerbated immune response and improvement in anti-tumor immunity (detection and elimination), by both direct and cross-reactions, had a fundamental role in this surprising chordoma regression with a reduction of 98.9% in the tumor volume. As T-cells are clearly involved in the COVID-19 immune response and were found infiltrating the tumor, we concluded that this cell must represent a critical pathway to this event. Fever and cross-reaction with brachyury may also have enhanced this process. As the tumoral piece resected in surgery was too small, it was not possible to perform additional analysis to elicit the interactions and define the specific mechanism that led to this regression. We encourage further immune analysis in patients with chordoma and confirmed COVID-19 in order to gain a better understanding of the SARS-CoV-2 and chordoma interaction.

## Figures and Tables

**Figure 1 viruses-17-00010-f001:**
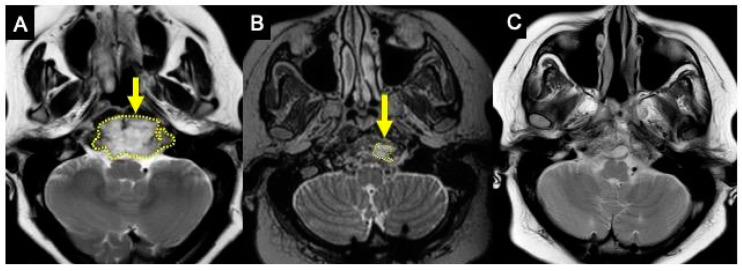
(**A**) Pre-COVID-19 T2W MRI in axial plane. Arrow points to the tumor. Dotted line at the limits of the tumor. Lesion measure: 6.5 × 3.5 × 4.5 cm. (**B**) Post-COVID-19 and Pre-Resection T2W MRI in axial plane. Arrow points to the remnant tumor. Dotted line at the limits of the tumor. Le-sion measures: 1.6 × 0.7 × 1 cm. (**C**) Post-Resection T2W MRI in axial plane showing complete re-section.

**Figure 2 viruses-17-00010-f002:**
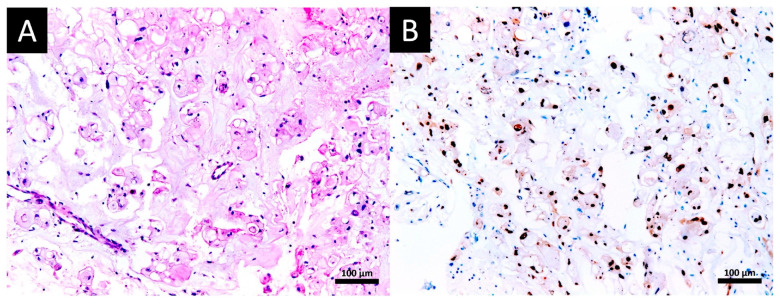
(**A**) H&E stain of small groups of tumor cells with a clear cytoplasm in a hyaline–myxoid matrix. (**B**) Positive brachyury immunohistochemical stain (marked in brown) and negative stain of abundant stromal inflammatory cells (marked in blue).

**Figure 3 viruses-17-00010-f003:**
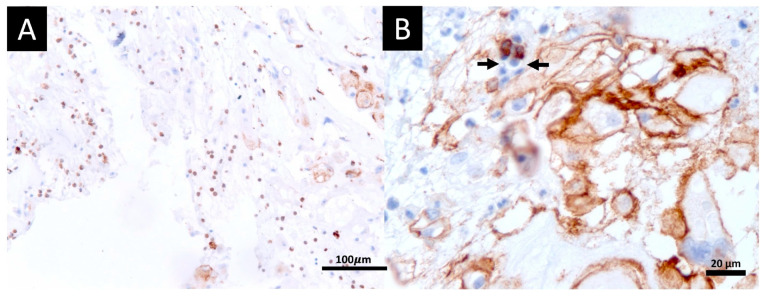
(**A**) Positive CD68 immunohistochemistry stain (marked in brown) showing tumor-infiltrating macrophages. (**B**) Positive CD56 immunohistochemical stain (marked in brown) in chordoma cells and negative stain of abundant stromal inflammatory cells (marked in blue). Two positive CD56 lymphocytes (arrows) are shown, which may be NK cells.

**Figure 4 viruses-17-00010-f004:**
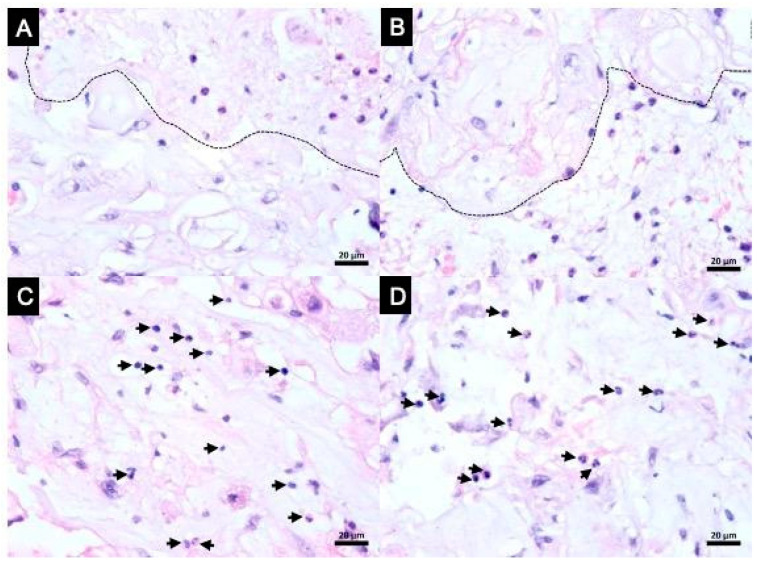
(**A**) H&E stain showing dotted line that separates the necrotic area with tumoral cells undergoing karyolysis under the line and the adjacent infiltrated area of tumoral inflammatory cells above the line. (**B**) H&E stain showing dotted line that separates the necrotic area with eosinophilic cytoplasm above the line and adjacent infiltrated area of tumoral inflammatory cells under the line. (**C**) H&E stain showing abundant tumor-infiltrating inflammatory cells (arrows). (**D**) Abundant tumor-infiltrating inflammatory cells (arrows).

**Figure 5 viruses-17-00010-f005:**
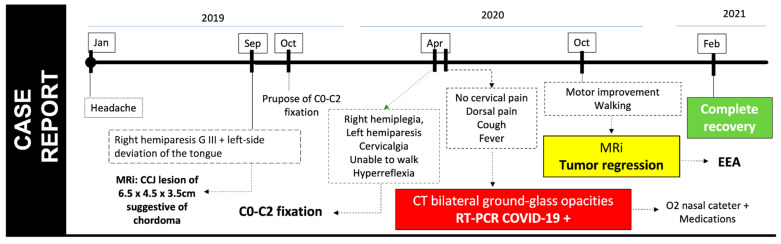
Case summary in timeline.

**Figure 6 viruses-17-00010-f006:**
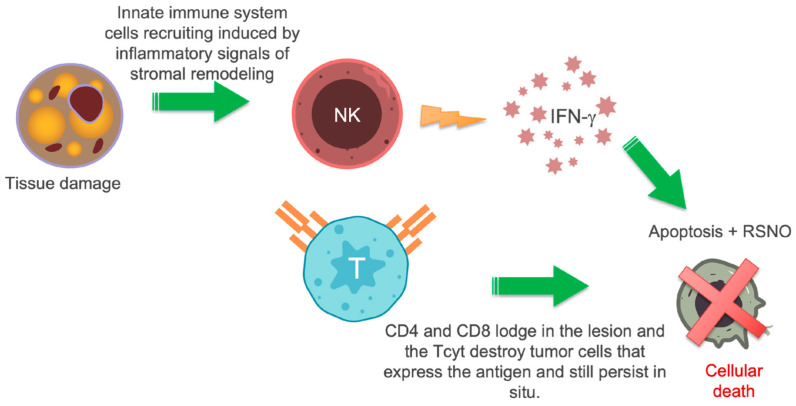
Tumor immunoediting stage of elimination. RSNO: reactive species of nitrogen and oxygen. Tcyt: cytotoxic T-lymphocyte.

**Figure 7 viruses-17-00010-f007:**
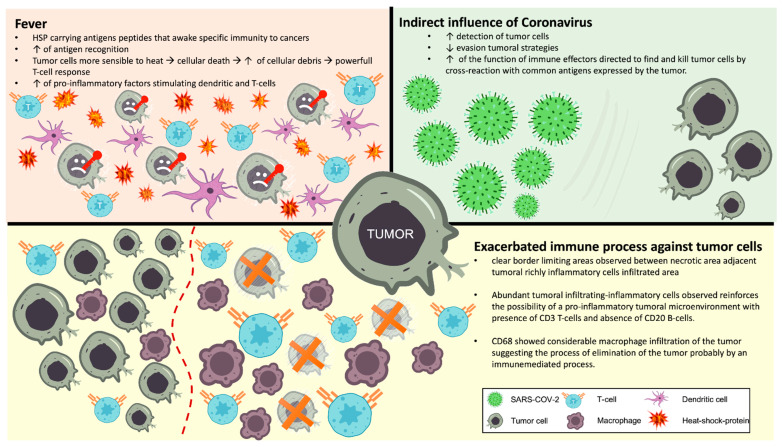
Possible mechanisms related to tumor regression: fever, indirect influence of the SARS-CoV-2 and exacerbated immune process against tumor cells.

**Table 1 viruses-17-00010-t001:** Summarized data of reported cases of chordoma spontaneous regression.

	Radl et al., 2005 [4]	Bander et al., 2020 [5]	Passeri et al., 2020 [6]	Vellutini et al., 2021 [7]	This Case
Location	Spine (C2)	Clivus	Clivus	Clivus	Clivus, C1–C2
Gender	Male	Female	Male	Female	Female
Symptoms	Neck pain, paresthesia, urinary and bowel disfunction, hyperreflexia	Stabbing headache	No.	Headache	Headache, neck pain, right hemiplegia, tongue palsy
Measure pre-regression	2.5 × 2 × 4 cm	3.1 × 1.9 × 3 cm (8.84 cm^3^)	36.9 cm^3^	-	6.5 × 3.5 × 4.5 cm (102.37 cm^3^)
Measure post-regression	-	2.3 × 2.1 × 1.4 cm (3.38 cm^3^)	24.7 cm^3^	-	1.6 × 0.7 × 1 cm (1.12 cm^3^)
Medications used	Dexamethasone, ciprofloxacin	Gabapentin, famotidine, loratadine, hydralazine, atenolol, nitroglycerin patch, vitamin B2, B6, B12. Complex of resveratrol, E-alpha-lipoic acid, hyaluronic acid, astaxanthin, echinacea, badger fat/oil tablets, sea buckhorn oil.	Corticosteroids, clarithromycin, doxycicline	-	Dexamethasone, meropenem, amikacin, ceftriaxone, clindamycin, azithromycin, hydroxycloroquine, ivermectin, prednisone, zinc and vitamin d, gabapentin, amitriptyline
Associated infection	Yes. Local abcess	No.	Yes. Inflammatory bullous dermatosis	No	COVID-19
Agent	Escherichia coli	-	Mycobacterium marinum	-	SARS-CoV-2
Pre-regression surgery	2. Biopsies	No	No	No	Fixation C0-C1-C2
Post-regression treatment	No. Complete regression	Ressection	Wait-and-see	Ressection	Ressection
PO fever	Yes	No	-	-	Yes
Histopathology	Chordoma	Chordoma	-	Chordoma	Chordoma
Brachyury	-	Negative	-	Positive	Positive

## Data Availability

Data are contained within the article.

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
