# Peer review of "Chordoma Spontaneous Regression After COVID-19"

_viruses, 2024, doi:10.3390/v17010010_

Round 1

Reviewer 1 Report

Comments and Suggestions for Authors

While this case report is intriguing and provides an unusual observation, it could benefit from additional context and analysis to strengthen its conclusions. The manuscript would be improved by a more comprehensive discussion of potential mechanisms, limitations, and alternative explanations. Additionally, including references to similar immune responses observed in other types of tumors post-COVID-19 infection or infection with other viruses would add depth.

If the authors could add further details on immune markers, expand the discussion on COVID-19's immunomodulatory effects, and address limitations, this case report could make a valuable contribution to the understanding of SARS-CoV-2’s effects on tumors, especially rare types like chordoma.

Author Response

Comment 1: 
While this case report is intriguing and provides an unusual observation, it could benefit from additional context and analysis to strengthen its conclusions. The manuscript would be improved by a more comprehensive discussion of potential mechanisms, limitations, and alternative explanations. Additionally, including references to similar immune responses observed in other types of tumors post-COVID-19 infection or infection with other viruses would add depth.
If the authors could add further details on immune markers, expand the discussion on COVID-19's immunomodulatory effects, and address limitations, this case report could make a valuable contribution to the understanding of SARS-CoV-2’s effects on tumors, especially rare types like chordoma.
Response 1: We expand this discussion correlating with other examples of solid tumors.  

Reviewer 2 Report

Comments and Suggestions for Authors

 da Silva Junior presented a case report of a female patient who had Chordoma and underwent surgery. Chordoma is a difficult type of tumor that has low survival (40%) unless surgery (65%). The patient in this case infected with SARS-CoV-2 after surgery and after 6 months she had significant regression of the tumor. This is an interesting case and needs to be reported.

However, there are some concerns that should be addressed.

1.       Author presented the survival percentage of chordoma. They should also mention the occurrence of chordoma in general population.

2.       Why it was not considered that the patient was recovered by surgery itself and not by COVID-19? It appears that surgery was done before COVID-19 (line 87). After surgery, author did not show the MRI image with the size of tumor? Author should clearly mention the what kind of surgery/procedure (post-operative/line 17) she had underwent. Was the tumor removed? But the figure 1 legend (1C) indicates that the surgery was done after COVID-19.   The text is confusing.

3.       Line 15-16, when the tumor was confirmed with immunohistologically, why she was still a “suspect” of chordoma.

4.       Line 259, SARS-CoV-2 normally suppresses IFN production including IFN-γ. Ref19 is very earlier (2020) review when most of the SARS-CoV-2 studies were not available. So, the explanation of IFN-γ on tumor recovery should be avoided.

5.       There is no patient consent mentioned. Authors must mention the ethical and patient consent for the study.

Author Response

da Silva Junior presented a case report of a female patient who had Chordoma and underwent surgery. Chordoma is a difficult type of tumor that has low survival (40%) unless surgery (65%). The patient in this case infected with SARS-CoV-2 after surgery and after 6 months she had significant regression of the tumor. This is an interesting case and needs to be reported.

However, there are some concerns that should be addressed.

Comment 1.       Author presented the survival percentage of chordoma. They should also mention the occurrence of chordoma in general population.

Response 1: We added the incidence information in the introduction. 

Comment 2.       Why it was not considered that the patient was recovered by surgery itself and not by COVID-19? It appears that surgery was done before COVID-19 (line 87). After surgery, author did not show the MRI image with the size of tumor? Author should clearly mention the what kind of surgery/procedure (post-operative/line 17) she had underwent. Was the tumor removed? But the figure 1 legend (1C) indicates that the surgery was done after COVID-19.   The text is confusing.

Response 2: As pointed out, on line 87, the caption of image 1. B is after COVID, but before resection (B: Post-COVID-19 Pre-Resection T2W MRI in axial plane...). Image 1. C is after resection. We therefore have: image 1. A Pre-COVID pre-resection; 1. B Post-COVID pre-resection; 1. C Post-resection. Also, line 17 details that only after COVID did she come back for resection. (Six months after COVID-19 she came back to endoscopic endonasal resection of the tumor...). The sequence was: symptoms -> MRi showing a large tumor -> occipitocervical fixation (arthrodesis) -> COVID-19 infection -> new MRI showing regression of tumor -> Resection of the regressed tumor -> histopathological and immunohistochemistry confirming diagnosis of chordoma.

Comment 3.       Line 15-16, when the tumor was confirmed with immunohistologically, why she was still a “suspect” of chordoma.

Response 3: We clarified the sentence. 

Comment 4.       Line 259, SARS-CoV-2 normally suppresses IFN production including IFN-γ. Ref19 is very earlier (2020) review when most of the SARS-CoV-2 studies were not available. So, the explanation of IFN-γ on tumor recovery should be avoided.

Response4: We correct the phrase, excluding the old reference and adding a new one. We exclude the previous explanation of IFN-γ on tumor recovery. 

Comment 5.       There is no patient consent mentioned. Authors must mention the ethical and patient consent for the study.

Response 5: We added this information before the author's contribution, and we attached the signed informed consent.

Round 2

Reviewer 1 Report

Comments and Suggestions for Authors

The authors fairly addressed my previous concerns.